# A Stress-Responsive NAC Transcription Factor from Tiger Lily (LlNAC2) Interacts with LlDREB1 and LlZHFD4 and Enhances Various Abiotic Stress Tolerance in Arabidopsis

**DOI:** 10.3390/ijms20133225

**Published:** 2019-06-30

**Authors:** Yubing Yong, Yue Zhang, Yingmin Lyu

**Affiliations:** Beijing Key Laboratory of Ornamental Germplasm Innovation and Molecular Breeding, National Engineering Research Center for Floriculture, College of Landscape Architecture, Beijing Forestry University, Beijing 100083, China

**Keywords:** NAC, DREB1, ZFHD4, interaction, abiotic stresses, lily

## Abstract

Our previous studies have indicated that a partial NAC domain protein gene is strongly up-regulated by cold stress (4 °C) in tiger lily (*Lilium lancifolium*). In this study, we cloned the full-length of this *NAC* gene, *LlNAC2*, to further investigate the function of *LlNAC2* in response to various abiotic stresses and the possible involvement in stress tolerance of the tiger lily plant. *LlNAC2* was noticeably induced by cold, drought, salt stresses, and abscisic acid (ABA) treatment. Promoter analysis showed that various stress-related cis-acting regulatory elements were located in the promoter of *LlNAC2*; and the promoter was sufficient to enhance activity of GUS protein under cold, salt stresses and ABA treatment. DREB1 (dehydration-responsive binding protein1) from tiger lily (LlDREB1) was proved to be able to bind to the promoter of *LlNAC2* by yeast one-hybrid (Y1H) assay. LlNAC2 was shown to physically interact with LlDREB1 and zinc finger-homeodomain ZFHD4 from the tiger lily (LlZFHD4) by bimolecular fluorescence complementation (BiFC) assay. Overexpressing *LlNAC2* in *Arabidopsis thaliana* showed ABA hypersensitivity and enhanced tolerance to cold, drought, and salt stresses. These findings indicated LlNAC2 is involved in both DREB/CBF-COR and ABA signaling pathways to regulate stress tolerance of the tiger lily.

## 1. Introduction

Plants have evolved a series of complex mechanisms to deal with environmental stressors, such as drought, high salinity, and extreme temperatures [1]. As crucial regulatory proteins, transcription factors (TFs) can specifically bind to cis-acting regulatory elements in the promoter of target genes to up or down-regulate their transcript levels [2,3]. Multiple types of TFs have shown to play important roles in plant responses to stresses, such as AP2/EREBP [4,5], MYB [6,7], bHLH [8], WRKY [9,10], zinc finger [11,12], and NAC [13,14] families. Thus, the identification and functional characterization of TFs contribute significantly to our understanding of the transcriptional regulatory networks in plants under abiotic stresses, which are indispensable parts in developing transgenic crops with enhanced tolerance to unfavorable growth conditions as well.

As one of the largest plant-specific TF groups, NAC (NAM, ATAF1,2 and CUC2) domain proteins have been reported to be essential regulators for plant response to abiotic stresses [15,16], which contain conserved N-terminal NAC DNA binding domains and variable C-terminal regions involved in transcriptional activation [17,18]. Most of the NAC genes were demonstrated to play positive roles in plant stress resistance [19]. For instance, overexpression of eight *NAC* genes (e.g., *OsNAC9*, *OsNAC10*) enhanced the drought and salinity tolerance significantly in rice (*Oryza sativa*) [1]. In wheat (*Triticum aestivum*), overexpression of *TaNAC2a*, *TaNAC67,* and *TaNAC47* enhanced multi-abiotic stress tolerances in transgenic plants [20,21]. In *Arabidopsis thaliana*, overexpression of *ANAC019*, *ANAC055,* and *ANAC072* genes enhanced plant tolerance to drought [22]; and *ATAF1* exhibited positive regulation of drought ABA, salt, and oxidative tolerance in the transgenic plants [23]. Also, overexpressing the *TsNAC1* (*Thellungiella halophila*), pumpkin (*Cucurbita moschata*) *CmNAC1* [24], cotton (*Gossypium hirsutum*) *GhNAC2* [25], rose (*Rosa rugosa*) *RhNAC3* [26] genes showed increased tolerance to diverse abiotic stresses in transgenic plants. In contrast, only a few *NAC* genes were found to function as negative regulators of tolerance regulation to abiotic stress, such as *NAC016* [27] and *NTL4* [28] in Arabidopsis, and *ShNAC1* (*Solanum habrochaites*) [19] and *SlSRN1* (*Solanum lycopersicum*) in tomato [29].

On the other hand, the identification of C-repeat binding factor (*CBF*) genes has been known as one of the most important milestones in the cold signaling network elucidation [30]. CBFs (CBF1/2/3), also called dehydration-responsive binding protein 1 (DREB1B/1C/1A), are from the APETALA2/ethylene responsive factor (AP2/ERF) family, which can bind to C-repeat/dehydration-responsive elements (CRT/DREs) in the promoters of certain cold-responsive genes (*COR*) and regulate their expressions and functions [31,32,33]. Many findings have implied that CBF can function as a master regulator of the cold signaling pathway. Interestingly, some studies reported that some NAC TFs were also involved in the DREB/CBF-COR pathway. For instance, in *Pyrus betulifolia*, PbeNAC1 induces the expression of some stress-associated genes by interacting with PbeDREB1 and PbeDREB2A [30]. GmNAC20 can bind to the promoter of *GmDREB1A* and suppress the expression of *GmDREB1C* to regulate *COR* genes and mediate cold tolerance in soybean (*Glycine max*) [33]. MaNAC1 can interact with MaCBF1 to mediate cold tolerance of banana (*Musa acuminata*) [34].

Lily is one of the most important flower crops cultivated worldwide. Most commercial cultivars of lily are sensitive to abiotic stresses, such as cold, drought, and salt. However, *Lilium lancifolium*, also called tiger lily, is one of the most widely distributed wild lilies in Asia which has strong abiotic stress resistance [35]. It suggests that tiger lily may have distinctive molecular mechanisms of abiotic stress resistance [36]. In our previous study, the unigene contig16924 coding for a partial NAC domain protein was found to be strongly up-regulated by low temperature in the tiger lily [36,37]. Thus, we characterized this *NAC* gene, *LlNAC2* in this report. *LlNAC2* was induced not only by cold stress but also by drought, salt, and exogenous ABA treatments. We also found the activity of *LlNAC2* promoter can be induced by cold, salt, and exogenous ABA treatments. In addition, A DREB1 from tiger lily (LlDREB1) was proved to bind to *LlNAC2* promoter, while LlNAC2 can physically interact with LlDREB1 and zinc finger-homeodomain ZFHD4 from tiger lily (LlZFHD4). Furthermore, the transgenic plants of *LlNAC2* showed good tolerance under cold, drought, and salt treatments, and activated the expression of many stress-responsive genes in Arabidopsis. Together, our results indicate that LlNAC2 plays a positive role in plant stress tolerance and can be a candidate gene utilized in transgenic breeding to enhance the abiotic stress tolerance of other crops.

## 2. Results

### 2.1. Cloning and Sequence Analysis of LlNAC2 Gene

*LlNAC2* gene contained a complete open reading frame (ORF) of 906 bp, with 74 bp 5′ UTR, 231 bp 3′ UTR. It encodes a polypeptide protein of 302 amino acids with a calculated molecular mass of 81.11 kDa and an isoelectric point of 5.03. Amino acid analysis revealed that the LlNAC2 contained a diverse activation domain in C-terminal, and a conserved NAC domain in the N-terminal region which can be divided into five subdomains, namely A to E (Figure 1a). A phylogenetic tree based on the sequences of LlNAC2 and some other known stress-responsive NACs was constructed, which revealed that LlNAC2 was clustered closely to rice ONAC048 and Arabidopsis ATAF1, with 65% and 63% of identity (Figure 1b). We also BLAST the DNA sequence of *LlNAC2* to the whole-genome sequence of *Arabidopsis thaliana* using The Arabidopsis Information Resource (TAIR). The BLAST result showed that the highest score (bits) significant alignment of *LlNAC2* was *AT1G01720.1* (*ATAF1*), which was located in the No. 1 chromosome (Appendix A).

### 2.2. Subcellular Localization and Transactivation Assay of LlNAC2

The GFP-LlNAC2 fusion construct and the GFP control in pBI121-GFP vector driven by CaMV35S promoter were transiently expressed in tobacco epidermal cells and visualized under a laser scanning confocal microscope to determine the subcellular localization of LlNAC2. Results showed the fluorescence signal from GFP alone was widely distributed throughout the cells, whereas the GFP-LlNAC2 fusion protein fluorescence signal was mainly detected in the nucleus (Figure 2a), which demonstrated that LlNAC2 is a nuclear protein.

The entire coding region, N-terminal, and C-terminal domain coding sequence were inserted into the pGBDKT7 vector to investigate the transcriptional activity of LlNAC2 protein. The transactivation results showed that all transformed yeast cells grew well on SD/-Trp medium (Figure 2b). The yeast strain containing the full-length LlNAC2 (LlNAC2-A) and the C-terminus of LlNAC2 (LlNAC2-C) could grow well on the selection medium SD/-Trp/-His/-Ade, while the cells with the N-terminus of LlNAC2 (LlNAC2-N) and pGBDKT7 empty vector could not grow normally (Figure 2b). Furthermore, the yeast cells that grew well on the SD/-Trp/-His-x-α-gal medium appeared blue in the presence of α-galactosidase (Figure 2b). These results indicated that LlNAC2 is a transcriptional activator, and its transactivation domain locates in the C-terminal region.

### 2.3. LlNAC2 is Induced by Multiple Stresses and ABA

The qRT-PCR analysis revealed that the *LlNAC2* gene has relative high expression level in bulb, while its expression level was low in leaf and stem (Figure 3a). Since *LlNAC2* was identified from cold-treated RNA-seq data [36], we also analyzed its expression levels under cold stress in stems, roots, and bulbs (Figure 3b). The results showed that *LlNAC2* was up-regulated in root and stem after 2 h at 4 °C treatment. However, at the same time, the expression of *LlNAC2* was strongly down-regulated and then up-regulated in bulb after 16 h at 4 °C treatment. Thus, we suppose that *LlNAC2* might play a special role in cold resistance of bulb in the tiger lily. Under ABA treatment, the expression of *LlNAC2* was significantly and rapidly up-regulated within 2 h, leading to a three-fold to four-fold increase, and then it was induced specifically again at 24 h (Figure 3c). However, treatment of tiger lily plants with cold and drought stress conditions induced the expression of *LlNAC2* until 24 h, leading to a five-fold to six-fold increase (Figure 3c,f). Similarly, salt treatment also induced the expression of *LlNAC2* slowly, showing a three-fold to five-fold increase during 12–24 h after treatment (Figure 3e). These results showed that *LlNAC2* is a stress-responsive *NAC* gene in the tiger lily.

The 1527 bp upstream of ATG start codon *LlNAC2* promoter sequence was also analyzed. Sequences of various putative stress-related and hormone-responsive cis-acting regulatory elements were identified in the *LlNAC2* promoter, including DRE/CRT (dehydration responsive element/C-repeat element) motif, LTRE (low-temperature-responsive element), MYCRS (MYC recognition site), MYBRS (MYB recognition site), ABRE (ABA responsive element), ARF (auxin response factor binding site), TGA-element and CGTCA-motif (Figure 3g, Appendix A). These cis-acting regulatory elements may be responsible for the induce expression of *LlNAC2* under stresses.

### 2.4. Promoter Activity of LlNAC2 Is Induced by Cold, Salt and ABA Treatments

To further clarify the regulatory mechanism underlying the stress-inducible expression of *LlNAC2*, the full-length *LlNAC2* promoter (proLlNAC2) was cloned and used to drive the *GUS* expression in Arabidopsis. The results showed that the untreated and drought-treated proLlNAC2 transformant (proLlNAC2-trans) Arabidopsis showed apparent lighter blue than that of the CaMV35S transformant (CaMV35S-trans) Arabidopsis plants (Figure 4a). However, the cold, salt, and ABA-treated proLlNAC2-trans showed darker blue than the untreated proLlNAC2-trans and the CaMV35S-trans Arabidopsis (Figure 4a), suggesting the expression of the *GUS* gene was increased after cold, salt stresses and ABA treatment, and the *GUS* transcript level driven by *LlNAC2* promoter is higher than that driven by CaMV35S. To further confirm these findings, *GUS* expression from the *LlNAC2* promoter was detected by qRT-PCR and fluorometric GUS assay. The expression of *GUS* gene increased significantly after 6 h or 12 h cold, salt, or ABA treatments in proLlNAC2-trans Arabidopsis plants (Figure 4b). From the GUS enzyme activity assay, we also found the same phenomenon as that shown in GUS histochemical staining results. Thus, the *LlNAC2* promoter mediated GUS activity increased under cold, salt, and ABA treatments.

### 2.5. LlDREB1 Can Bind to the Promoter of LlNAC2

DREBs were known to play a vital role in plant response to abiotic stresses, and some NAC TFs were found to be involved in the DREB/CBF-COR pathway in many studies [30,33,34,38] which inspired us to further explore the NAC-DREB/CBF-COR signaling cascade. In our previous study, a novel *DREB1* gene from the tiger lily (LlDREB1, NCBI accession No. KJ467618) induced by cold stress was identified [35]. Thus, we performed the Y1H assay to explore whether there is an interaction between the LlDREB1 protein and *LlNAC2* promoter. The minimal inhibitory concentration of Aureobasidin A (AbA) for bait yeast strains was found to be 150 ng/mL (Appendix A). Yeast cells transformed with pGADT7-LlDREB1 and pAbAi-CRT/DREs grew well on SD/Leu plates with 150 ng/mL and 200 ng/mL AbA (Figure 5a). This suggested that LlDREB1 possessed the ability to bind CRT/DREs as general CBF/DREB1 TFs. The fragment (−487 to −312) of the *LlNAC2* promoter containing three CRT/DREs was cloned and its interaction with LlDREB1 was detected (Figure 5b); the result showed LlDREB1 could bind to this fragment, indicating LlDREB1 might be involved in the regulatory pathway of *LlNAC2*.

### 2.6. LlNAC2 Interacts with LlDREB1 or LlZFHD4

We have already found LlNAC2 was highly co-expressed with another cold-responsive TF zinc finger homeodomain protein LlZFHD4 [35]. To further explore the interactions between LlNAC2, LlZFHD4, and LlDREB1 proteins, BiFC assay was performed with a yellow fluorescent protein (YFP). YFP was observed in tobacco epidermal cells, transformed with vectors containing LlNAC2-pSPYNE/LlZFHD4-pSPYCE and LlNAC2-pSPYNE/LlDREB1-pSPYCE localized to the nuclei; while no YFP was detected in negative control pSPYNE173/pSPYCE (M) and LlZFHD4-pSPYNE/LlDREB1-pSPYCE (Figure 6). The result showed that LlNAC2 could interact with LlDREB1 and LlZFHD4.

### 2.7. Overexpression of LlNAC2 in Arabidopsis Enhance Tolerance to Cold and Drought Stresses

Among 12 independent homozygous *LlNAC2* transgenic lines, LlNAC2-5 (L5) and LlNAC2-6 (L6) with relatively high expression levels (Appendix A) were selected for further analysis.

To study the effect of *LlNAC2* overexpression on cold stress, *LlNAC2* transgenic lines and wild-type (WT) plants were grown in equal amounts of potting soil for 3 weeks under normal conditions, and cold stress was applied by being exposed to various freezing temperatures for 12 h. The results showed that all plants grew well under −2 °C and −4 °C treatment as same as under normal temperature 22 °C (Figure 7a). When the temperature was decreased to −6 °C, most of the WT plants were dead with a survival rate at around 19%, but over half of two transgenic plants survived (Figure 7a,c). At −8 °C, all WT plants were dead whereas the survival rate for transgenic plants was observed at 30%–35% (Figure 7a,c). In a further experiment, 3-week-old plants were treated at 4 °C for 3 h, and the relative electrolyte leakage and soluble sugars were measured after treatment. As a result, the transgenic plants showed lower electrolyte leakage and higher levels of soluble sugars relative to WT plants (Figure 7d,e).

Similarly, to study drought stress tolerance, WT plants showed visible symptoms of drought-induced damage and even death after withholding water for 30 days, while some transgenic plants remained green with expanded leaves (Figure 7b). When plants were rehydrated, about 80% of the transgenic plants recovered well, whereas the WT plants failed and ultimately died (Figure 7b,c). Additionally, after being treated with 16.1% PEG6000 (−0.5 MPa) for 3 h, transgenic plants showed lower electrolyte leakage and higher levels of soluble sugars compared to WT plants (Figure 7d,e). Also, the water-loss rates were found lower in transgenic plants after 3 h treatment (Figure 7f). These results suggest that the *LlNAC2* transgenic plants conferred tolerance to cold and drought stresses in some degree.

### 2.8. Overexpression of LlNAC2 in Arabidopsis Increases Seed Sensitivity to ABA and Tolerance to NaCl

Salt tolerance and ABA sensitivity of *LlNAC2* transgenic plants were also assessed. WT and two *LlNAC2* transgenic lines were placed on MS agar plates supplemented with 50 mM or 100 mM NaCl and 2 µM ABA. There were no differences in plant morphology between WT and LlNAC2 transgenic Arabidopsis on MS agar plates (Figure 8a,b). However, *LlNAC2* transgenic Arabidopsis displayed significantly higher germination ratios than WT plants following NaCl treatment according to both cotyledon greening and radicle protrusion (Figure 8a,c,d). In contrast, the germination ratio of transgenic plants seeds was remarkably lower than that of WT seeds in the MS medium containing 2 µM ABA according to cotyledon greening (Figure 8b,c). Therefore, we suggested that *LlNAC2* transgenic plants are more tolerant to salt stresses and more hypersensitive to ABA than WT plants.

### 2.9. Altered Expression of Stress-Related Genes in LlNAC2 Transgenic Arabidopsis Plants

*LlNAC2* transgenic Arabidopsis plants showed enhanced tolerance to cold, drought, and salt stresses. Thus, we detected the transcript levels of some stress-responsive genes from Arabidopsis in the transgenic plants, including *AtRD20*, *AtCOR47*, *AtRD29A*, *AtRD29B*, *AtLEA14, AtGolS1*, *AtAPX2,* and *AtGSTF6* genes (NCBI accession numbers are shown in Appendix A). The qRT-PCR results showed higher transcripts of these genes accumulated in *LlNAC2* transgenic plants compared to WT plants (Figure 9). We suppose the activated expression of these stress-responsive genes may contribute to the stronger stress tolerance in *LlNAC2* transgenic plants.

## 3. Discussion

In this study, we identified a novel stress-related NAC TF gene, *LlNAC2*, from tiger lily. Sequence analysis shows that LlNAC2 has a highly conserved sequence in the N-terminal region and has high sequence identity with ATAF1 from Arabidopsis. Like most transcription factors, LlNAC2 protein is localized in the nucleus with transactivation activity in the C-terminal region. Our previous transcriptome data analysis identified a unigene contig16924 coding for partial LlNAC2, which showed significant changes in expression in the tiger lily under cold stress [35,36]. Here, we further confirmed that *LlNAC2* not only participates in cold stress response, but also responds to drought and salt stress, and its expression is sensitive to ABA signaling molecules as well. Also, the *LlNAC2* promoter is shown to be a stress-inducible promoter, and the GUS activity driven by the *LlNAC2* promoter is significantly higher than that driven by the CaMV35S promoter. Given that using a stress-inducible promoter to drive transgene expression can not only result in remarkable gains in stress tolerance, but also avoid impacting important traits in crops negatively [39], the *LlNAC2* promoter can be a candidate stress-inducible promoter utilized in transgenic breeding to enhance the stress tolerance of crops.

Many NAC domain proteins from different species were shown to function positively in regulation of plant stress tolerance [1,21,40,41]. We also found tolerance to cold, drought, and salt stresses were likely conferred by overexpressing *LlNAC2* in Arabidopsis. At the morphological level, compared to WT plants, *LlNAC2* transgenic lines showed a higher seed germination ratio under salt stress condition, and performed better growth performance and higher survival rate under drought and cold stress conditions. At the physiological level, the lower electrolyte leakage amount and higher soluble sugar level were observed in transgenic plants after dehydration and chilling treatment. Furthermore, the leaf water-loss ratios of the *LlNAC2* transgenic plants were lower in transgenic plants than in WT plants. These physiological indices changes likely resulted in enhancing tolerance to stresses at the physiological level, which were consistent with previous reports about NAC TFs in maize, rice, and rose [21,26,42]. At the gene transcription level, the *LlNAC2* gene may improve stress tolerance by regulating downstream stress-responsive genes. The results showed that higher transcript levels of 8 picked stress-responsive genes accumulated in *LlNAC2* transgenic plants. More importantly, putative NAC binding cis-elements were also reported to localize in the promoter sequences of these genes [27,43,44], indicating these genes might be transcriptionally regulated directly by LlNAC2.

On the other hand, we found overexpression of *LlNAC2* resulted in improved ABA sensitivity, which was also observed on many NAC TFs from other species, such as GmNAC20, MlNAC5, AtATAF1, ONAC022, and TaNAC47 [1,21,23,42,45]. Promoter analysis showed that there are two ABREs (YACGTGGC, Y = C/T) and eight core DPBF (Dc3 promoter-binding factor) (ACACNNG) cis-acting elements in *LlNAC2* promoter. ABREs can be recognized by ABRE-binding factors (ABRE/ABFs), which are involved in ABRE-dependent ABA signaling pathway in response to drought stress [46]; the core DPBF (Dc3 Promoter-Binding Factor) (ACACNNG) elements were also known to participate in ABA responsiveness [47,48]. The presence of these cis-elements might explain why *LlNAC2* was induced more quickly in response to ABA treatment compared with other stresses treatment in the tiger lily. Given that most of the picked genes above were also reported to be ABA-responsive genes, these results suggest that the LlNAC2 may go through the more efficient ABA signaling pathway to enhance cold, drought, and salt stress tolerance in transgenic plants.

Except for the ABA signaling pathway, NAC domain proteins have received much attention as regulators in various stress signaling pathways [49], in which some studies showed some NAC TFs were also involved in the DREB/CBF-COR pathway [1,30,33,34]. CBF/DREB1 is known to activate stress-responsive genes under abiotic stress conditions by binding to specific cis-elements such as DRE/CRTs present in their promoters [4,50]. In the promoter region of *LlNAC2*, we found there are one LTRE (CCGAC) and three CRT/DREs (GC/TCGAC) elements, which are known to be necessary and sufficient for gene transcription under cold stress [51,52,53,54]. Furthermore, a novel *DREB1* gene from the tiger lily (*LlDREB1*) induced by cold stress was identified in our earlier study [35]. We thus further analyzed the interactions between LlDREB1 and LlNAC2. Y1H assay confirmed that LlDREB1 could bind to the *LlNAC2* promoter. It provides a direct evidence for *LlNAC2* as a novel target of LlDREB1. Interestingly, BiFC assay showed that the LlNAC2 protein can interact with the LlDREB1 protein in tobacco epidermal cells. These findings indicate that the *LlNAC2* gene might be regulated by LlDREB1 TF under abiotic stress conditions and LlNAC2 TF can in turn interact with LlDREB1 to more efficiently regulate the expression of downstream stress-related genes. It may also explain why *LlNAC2* was induced more slowly but more strongly in response to cold compared with salt, drought, and ABA treatments in the tiger lily. However, in soybean, GmNAC20 was found to be an upstream protein of GmDREB1A which can directly bind to the promoter of *GmDREB1A* [1]. Thus, we suppose that there are various ways for NAC TFs participating in the DREB/CBF-COR pathway, and LlNAC2 is involved in DREB/CBF-COR pathway, to mediate stress tolerance of the tiger lily.

In Arabidopsis, an interaction between stress-inducible zinc finger homeodomain ZFHD1 and NAC TFs was detected in the report of Lam-Son Phan Tran [55]. It showed co-overexpression of *ZFHD1* and *NAC* genes, enhanced expression of *ERD1* in both Arabidopsis T87 protoplasts, and transgenic Arabidopsis plants. Also, in our previous study, *LlNAC2* was highly co-expressed with another cold-responsive TF, the zinc finger homeodomain protein 4 LlZFHD4 [36]; and here we confirmed that the LlNAC2 protein can physically interact with the LlZFHD4 protein by BiFC assay. However, further analysis is needed to confirm whether this interaction functions the same as AtZFHD1 and AtNAC TFs in Arabidopsis.

In conclusion, LlNAC2 is a nucleus-localized transcriptional activator which is regulated by cold, drought, and salt stresses and sensitive to ABA. The *LlNAC2* promoter is a stress-inducible promoter which showed higher promoter activity than CaMV35S under cold, salt, and ABA conditions. Overexpressing *LlNAC2* in Arabidopsis showed ABA hypersensitivity and enhancing tolerance of transgenic plants to freezing, dehydration, and salt conditions. We demonstrated that the *LlNAC2* gene is not only a novel direct target of LlDREB1, but the LlNAC2 protein can also interact with the LlDREB1 protein. These data indicate LlNAC2 is involved in both DREB/CBF-COR and ABA-dependent pathways to mediate stress tolerance of the tiger lily (Figure 10). Moreover, the protein interaction of the co-expressed LlNAC2 and LlZFHD4 gene was confirmed, providing another pathway involving regulator LlNAC2. Therefore, our future efforts will be focused on investigating the role of these interactive proteins in regulating expression and modulating the function of LlNAC2 under various stress conditions.

## 4. Materials and Methods

### 4.1. Plant Materials and Abiotic Stress Treatments

The wild tiger lily (*Lilium lancifolium*) was used as experimental material in this study. The seedlings preparation method was described in our previous study [36]. The bulbs of the tiger lily were cleaned, disinfected, and then stored at 4 °C; in March, the bulbs were box-cultivated in a greenhouse (116.3° E, 40.0° N) under controlled conditions. The model plant *Arabidopsis thaliana* Columbia-0 (Col-0) was selected for the transgenic study of *LlNAC2*. Arabidopsis plants were grown in 8 cm × 8 cm plastic pots containing a 1:1 mixture of sterile peat soil and vermiculite under controlled conditions (22/16 °C, 16 h light/8 h dark, 65% relative humidity, and 1000 lx light intensity). Seeds of *Nicotiana benthamiana* were planted and cultured under the same conditions.

For the expression analysis of *LlNAC2* in response to abiotic stress and ABA treatment, 8-week-old tiger lily seedlings were treated with 4 °C, 16.1% PEG6000 (−0.5 MPa), 100 mM NaCl and 100 µM exogenous ABA for 0, 1, 3, 6, 12, and 24 h, respectively. Leaf samples were collected and immediately frozen with liquid nitrogen and stored at −80 °C for RNA isolation.

### 4.2. Full-Length cDNA Cloning and Sequence Analysis of LlNAC2

The 3′-complete sequence cDNA of *LlNAC2* gene was obtained from the transcriptome data of cold-treated tiger lily leaves in our laboratory. Using the SMARTer RACE 5′/3′ Kit (Clontech, United States), we performed a 5′-rapid amplification of cDNA ends (5′-RACE) of *LlNAC2* according to the manufacturer’s protocol. Two reverse primer pairs (Appendix A) were designed to amplify the 5′-complete sequence cDNA of *LlNAC2*. The 3′- and 5′-sequences were assembled by DNAMAN (version 7), resulting in the full-length cDNA of *LlNAC2* gene. The coding sequence of *LlNAC2* was amplified and cloned into pEASYT1-Blunt vector (TransGen Biotech, Beijing, China). Plasmid pEASYT1-LlNAC2 was used as a template for all experiments after sequencing. Multiple sequence alignments were performed using DNAMAN (version 7). Phylogenetic tree analysis was performed using the neighbor-joining method in MEGA5 software with 1000 replications. The NCBI accession numbers of genes used in multiple sequence alignments and phylogenetic tree analysis are shown in Appendix A. The theoretical molecular weight and isoelectric point were calculated using ExPASy (http://expasy.org/tools/protparam.html). The NAC domain region was identified with SCANPROSITE (http://www.expasy.ch/tools/scanprosite/).

### 4.3. RNA Isolation and Quantitative Real-Time PCR Analysis

Total RNA was isolated using an RNAisomate RNA Easyspin isolation system (Aidlab Biotech, Beijing, China). First-strand cDNA synthesis was performed using Prime Script II 1st strand cDNA Synthesis Kit (Takara, Shiga, Japan) according to the manufacturer’s instructions. The qRT-PCR was performed using a Bio-Rad/CFX Connect™ Real-Time PCR Detection System (Bio-Rad, CA, USA) with SYBR^®^ qPCR mix (Takara, Shiga, Japan) according to the manufacturer’s protocol. Relative mRNA content was calculated using the 2−^△△^*^C^*^t^ method against the internal reference gene encoding tiger lily tonoplast intrinsic protein 1 (LlTIP1) [35] and Arabidopsis *Atactin* gene (NCBI accession No. NM_112764). The primers used in this study were designed with Primer Premier 5 and are listed in Appendix A. All reactions were performed in three biological replicates. Student’s t-test was performed for all statistical analysis in this study.

### 4.4. Promoter Cloning and Sequence Analysis

*LlNAC2* promoter sequence was cloned from tiger lily genomic DNA using Genome Walker Universal Kits (Clontech, United States). The cis-acting motifs present in the LlNAC2 promoter were predicted using the online search tool PLACE (http://www.dna.affrc.go.jp/PLACE/) database.

### 4.5. Subcellular Localization of LlNAC2

By using ClonExpressII One Step Cloning Kits (Vazyme, Piscataway, NJ, United States), full-length *LlNAC2* was inserted into vector pBI121-GFP at *Xho*I and *Sal*I sites. The pBI121-LlNAC2-GFP plasmid and the pBI121-GFP empty vector were transformed into *Agrobacterium tumefaciens* GV3101 and infiltrated separately into *N. benthamiana* leaves. After infiltration, the plants were grown in a growth room under controlled conditions (22/16 °C, 16 h light/8 h dark, 65% relative humidity, and 1000 lx light intensity) for 32 h. GFP fluorescence signals were excited at 488 nm and detected under Leica TCS SP8 Confocal Laser Scanning Platform (Leica SP8, Leica, America) using a 500–530 nm emission filter.

### 4.6. Transcription Activation Assay of LlNAC2 in Yeast 

To investigate the transcriptional activity of the LlNAC2 protein, the full-length coding sequence of *LlNAC2* and the sequence encoding the N-terminus (1–453 bp) and C-terminus (454–906 bp) were inserted into the *EcoR*I and *BamH*I sites of the pGBKT7 vector, resulting in plasmid pGBKT7-LlNAC2-A, pGBKT7-LlNAC2-N (1–151 aa), and pGBKT7-LlNAC2-C (152–302 aa). These plasmids were transformed into the yeast strain Y2HGold separately by using Quick Easy Yeast Transformation Mix (Clontech, United States). The transformed yeast cells were incubated on SD/-Trp, SD/-Trp/-His-Ade and SD/-Trp-His-x-α-gal plates [24].

### 4.7. Yeast One-Hybrid (Y1H) Assays

Y1H assay was carried out using the Matchmaker™ Gold Yeast One-Hybrid System (Clontech, United States). Three tandem copies of CRT/DRE were generated by oligonucleotide synthesis and cloned into the pAbAi (bait) vector (Clontech, United States). A fragment (−312 to −487) of the *LlNAC2* promoter was amplified by PCR, and also cloned into the pAbAi (bait) vector to generate pAbAi-NAC-CRT/DREs plasmid (shown in Figure 4B). Full-length *LlDREB1* (NCBI accession No.KJ467618) was amplified and inserted into pGADT7 (prey) vector (Clontech, United States) yielding plasmid pGADT7-LlDREB1. The bait plasmids were linearized and transformed into the yeast strain Y1HGold. Positive yeast cells were then transformed with pGADT7-LlDREB1 plasmid. The DNA–protein interaction was determined based on the growth ability of the co-transformants on SD/-Leu medium with Aureobasidin A (AbA).

### 4.8. BiFC Assay

Full-length *LlNAC2* and zinc finger homeodomain protein *LlZFHD4* (sequence shown in Appendix A) were cloned into the pSPYNE173 vector, *LlDREB1* and *LlZFHD4* were cloned into the pSPYCE(M) vector [56], respectively. Co-expression was executed on *N. benthamiana* leaves as described in subcellular localization assessments. After infiltration, the plants were grown under 24 h dark and then 16 h light/8 h dark for 32 h. YFPs were excited at 514 nm. Images were generated through Leica TCS SP8 Confocal Laser Scanning Platform using a 500–530 nm emission filter.

### 4.9. Generation of LlNAC2 Transgenic Arabidopsis

The *LlNAC2* open read frame (ORF) was cloned into the pBI121 vector under control of a CaMV35S promoter; the *LlNAC2* promoter region was inserted into the CaMV35S-GUS vector by replacing the CaMV35S promoter. The recombinant vectors and empty GUS vector were transformed into 5-week-old Arabidopsis ecotype Col-0 plants by the floral-dip method [57]. Transformed seeds were selected on MS medium containing 50 mg/L kanamycin. All transgenic lines were identified by RT-PCR; T3-generation homozygous lines were selected for gene functional analysis.

### 4.10. Histochemical Staining and Fluorometric GUS Assay

Histochemical staining and fluorometric GUS assay analysis for GUS activity was carried out as described before [58]. Transgenic *LlNAC2* Arabidopsis plants were treated with 4 °C, 16.1% PEG6000 (−0.5 MPa), 100 mM NaCl and 100 µM exogenous ABA for different durations before sampling. The leaves of stress-treated transgenic *LlNAC2* Arabidopsis were incubated in GUS reaction buffer (Huayueyang Biotech, Beijing, China). Photos of those stained samples were obtained by a Leica TL3000 Ergo microscope under white light. Leaves of stress-treated transgenic Arabidopsis were also used to examine the GUS gene expression level by qRT-PCR, and determine GUS enzyme activity.

### 4.11. Abiotic Stress Tolerance Assays and ABA Sensitivity Analysis

For cold and drought treatments, 3-week-old seedlings were kept at 4 °C for 3 h, then at −2, −4, −6 or 8 °C, respectively, for 12 h. After that, the plants were kept at 4 °C for 3 h before transferring to a normal condition at 22 °C [34]. For the drought treatment, the water intake of 3-week-old seedlings in water-saturated substrate was withheld for 30 days, followed by rehydrating the seedlings for 7 days [24].

For determining the salt tolerance and ABA sensitivity in transgenic plants, Arabidopsis seeds were cultivated on MS medium supplemented with 0 and 2 µM ABA or 50 and 100 mM NaCl, respectively, under continuous light at 22 °C in a growth chamber. The germination rate was scored on the 7th day after planting on the plates.

### 4.12. Measurements of Relative Electrolyte Leakage, Soluble Sugar, and Water Loss Rate

The relative electrolyte leakage, soluble sugar content, and water loss rate were evaluated following the method described previously [21,59]. The relative electrolyte leakage was evaluated by determining the relative conductivity of fresh leaves (100 mg) in solution using a conductivity detector. The anthrone-sulfuric acid colorimetry was used for determining the soluble sugar. The water loss rate was calculated related to the initial fresh weight of the leaf samples; the samples were placed on the lab bench (20–22 °C, humidity 45−60%) and weighed at designated time points. All the measurements were performed with ten plants in triplicate.

## Figures and Tables

**Figure 1 ijms-20-03225-f001:**
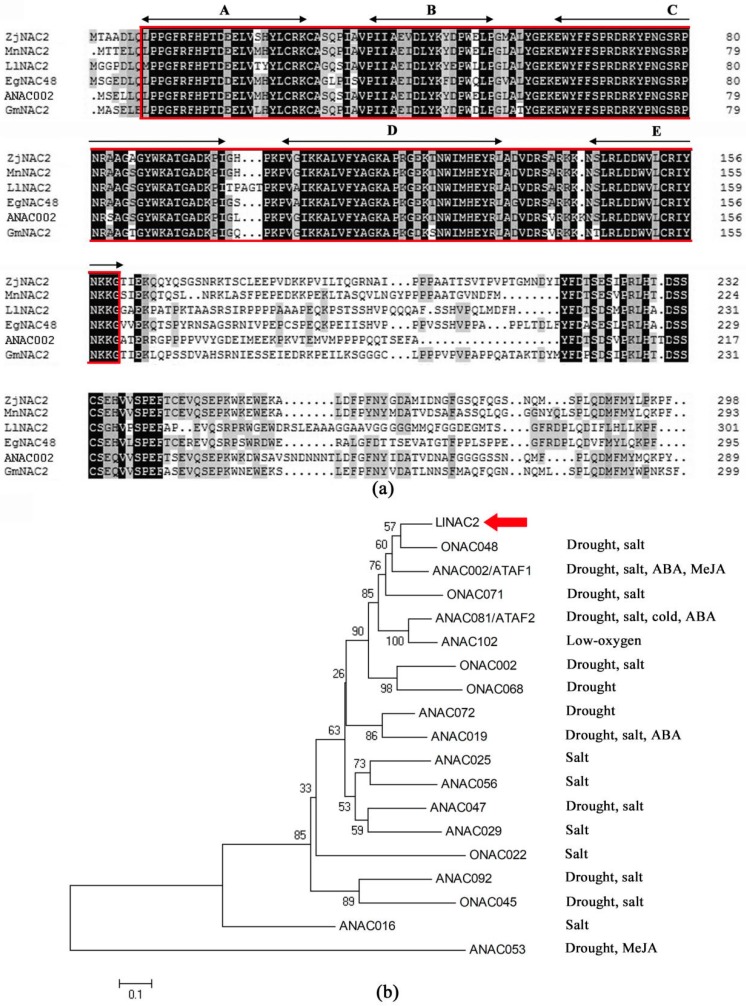
Characterization of LlNAC2. (**a**) Alignment of LlNAC2 with *Ziziphus jujuba* ZjNAC2, *Morus notabilis* MnNAC2, *Elaeis guineensis* EgNAC48, Arabidopsis ANAC002, and *Glycine max* GmNAC2. Black arrowed lines indicate the locations of the five highly conserved subdomains A–E. The conserved NAC domain is boxed and identical amino acids are shaded in black. (**b**) Phylogenetic tree analysis of LlNAC2 with other known stress-responsive NAC proteins. Protein sequences used in multiple sequence alignments and phylogenic tree analysis are shown in Appendix A.

**Figure 2 ijms-20-03225-f002:**
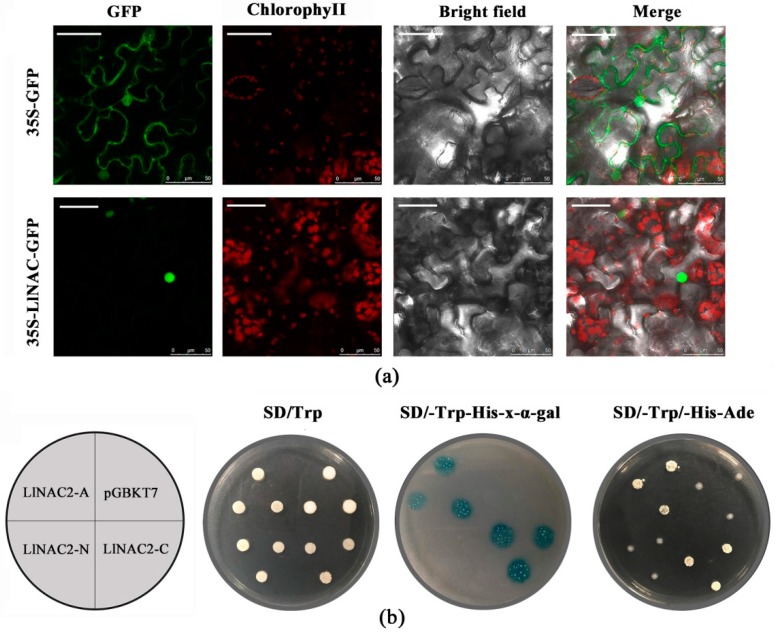
Nuclear localization and transactivation assay of LlNAC2. (**a**) 35S-GFP and 35S-LlNAC2-GFP fusion proteins were transiently expressed in tobacco leaves and observed under a laser scanning confocal microscope. GFP, chlorophyII, bright field, and merged images were taken (Scale bar, 50 μm). (**b**) Full-length protein (LlNAC2-A), N-terminal fragment (LlNAC2-N) and C-terminal fragment (LlNAC2-C) were inserted to the pGBKT7 vector and expressed in yeast strain Y2HGold. Transformed yeasts were dripped on the SD/-Trp, SD/-Trp-His-x-gal and SD/-Trp-His-Ade plates after being cultured for 3 days under 30 °C. The pGBDKT7 vector was used as a negative control.

**Figure 3 ijms-20-03225-f003:**
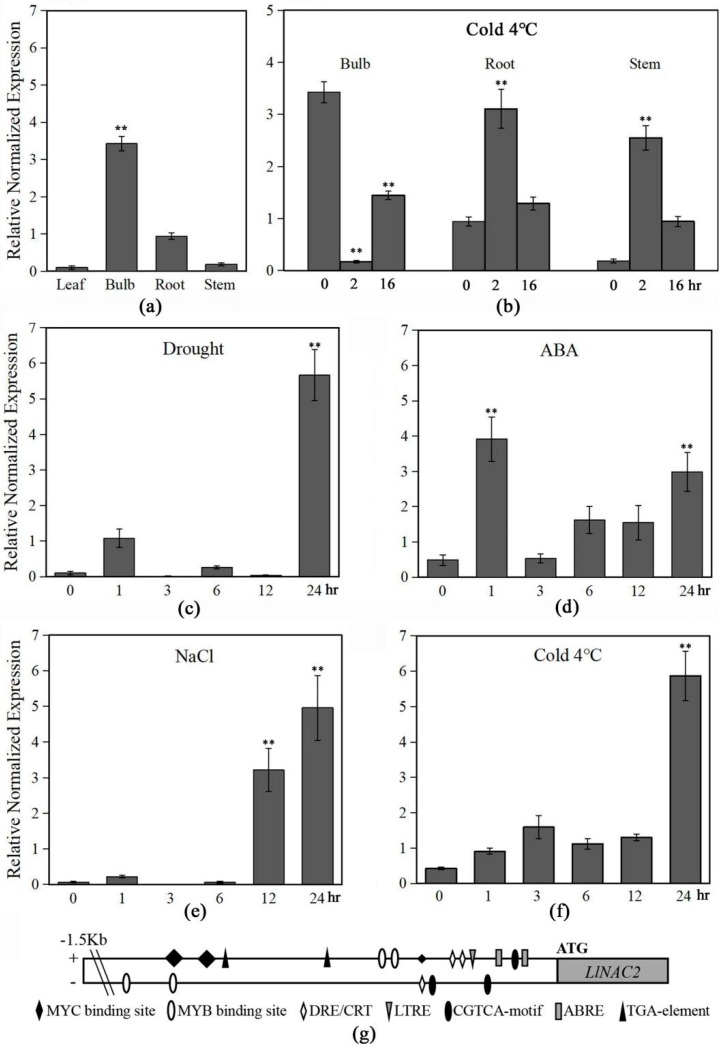
Expression patterns of *LlNAC2* in tiger lily seedlings under different stress treatments and the distribution of stress-related cis-elements in the *LlNAC2* promoter. Expression patterns of *LlNAC2* in leaves, bulbs, roots, and stems (**a**), and expression patterns of *LlNAC2* after cold treatments in leaves (**c**), bulbs, roots, and stems (**b**), and after ABA (**d**), NaCl (**e**), drought (**f**) treatments by qRT-PCR analysis. Transcript levels were normalized to *LlTIP1*. Three biological replications were performed. The bars show the SD. Asterisks indicate a significant difference ** *p* < 0.01 compared with the corresponding controls. (**g**) Distribution of major stress-related cis-elements in the promoter region of *LlNAC2* gene.

**Figure 4 ijms-20-03225-f004:**
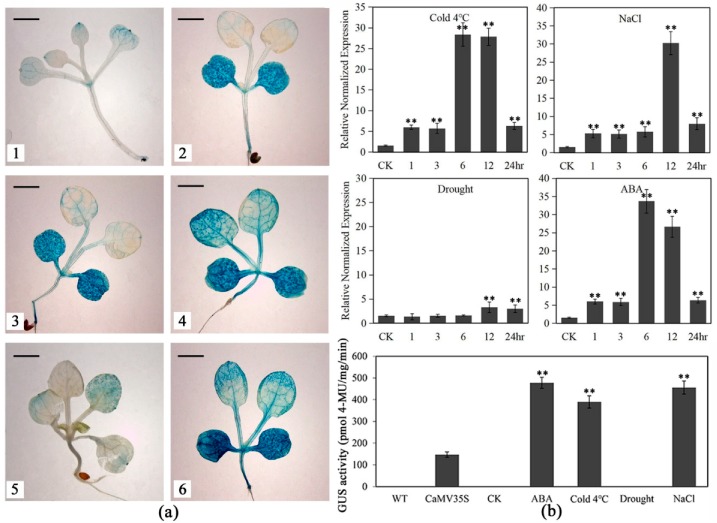
GUS activity mediated by *LlNAC**2* promoter in transgenic Arabidopsis plants. (**a**) Beta-glucuronidase (GUS) expression in untreated (1), cold (3), NaCl (4), drought (5), and ABA-treated (6) proLlNAC2-trans and CaMV35S-trans (2) Arabidopsis plants (Scale bar, 2 mm). (**b**) The *GUS* transcript levels and enzyme activity in the leaves of the transgenic Arabidopsis under cold (4 °C), salt, drought, and ABA treatments. The transgenic plants treated for 12 h under above treatments were used for fluorometric GUS assay. GUS activity from the CaMV35S-trans, untreated proLlNAC2-trans (CK), and wild type (WT) served as controls. Twelve transgenic lines were acquired. Three biological replications were performed. The bars show the SD. Asterisks indicate a significant difference (** *p* < 0.01) compared with the corresponding controls.

**Figure 5 ijms-20-03225-f005:**
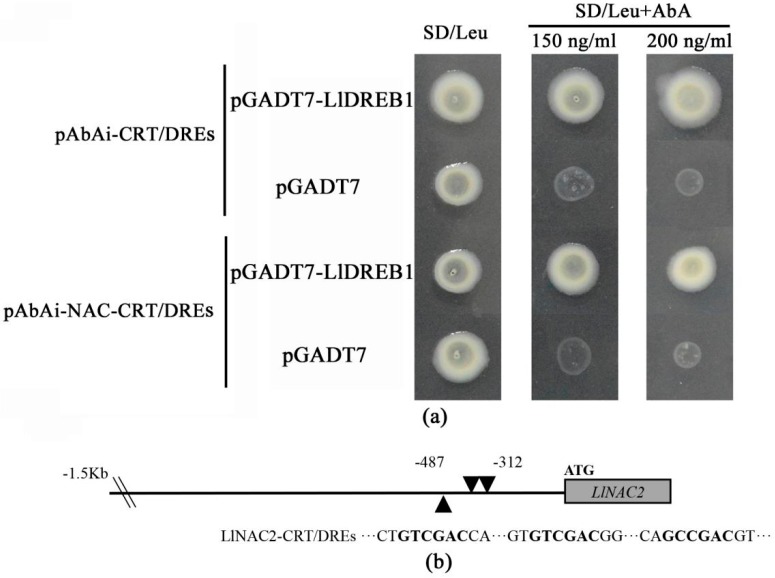
Yeast one-hybrid analysis of LlDREB1 binding to *LlNAC2* promoter. (**a**) Yeast one-hybrid analysis. Three tandem repeats of a CRT/DRE and the −487 to −312 region of the *LlNAC2* promoter were inserted in front of the reporter gene AUR1-C. Yeast strain Y1HGold was co-transformed with bait (pAbAi-CRT/DREs or pAbAi-NAC-CRT/DREs) and a prey (pGADT7 or pGADT7-LlDREB1) construct. Interaction between bait and prey was determined by cell growth on SD medium lacking Leu in the presence of 200 ng mL^−1^ AbA. (**b**) Schematic representation of CRT/DRE recognition elements in the *LlNAC2* promoter.

**Figure 6 ijms-20-03225-f006:**
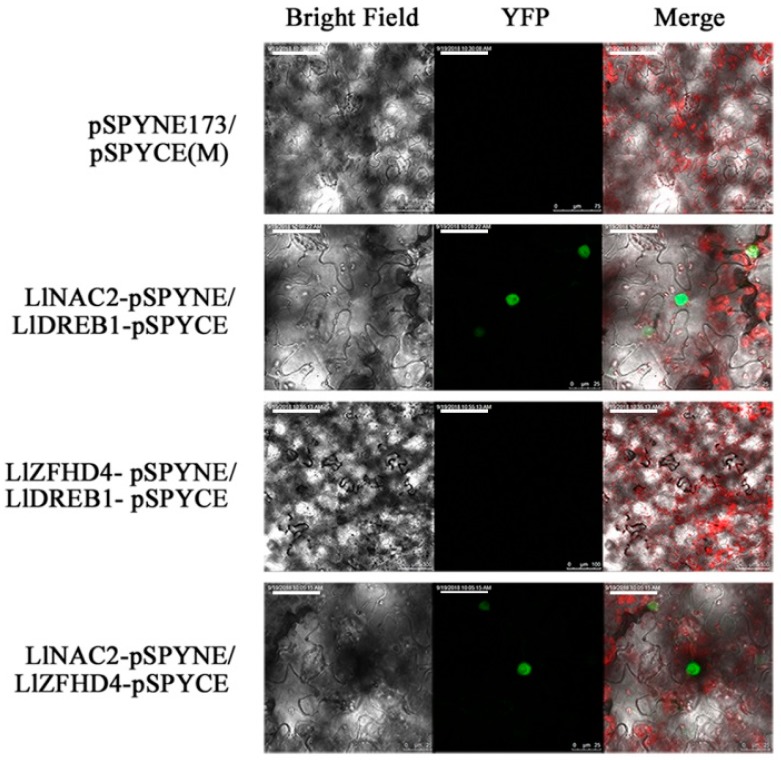
Bimolecular fluorescence complementation assay using tobacco epidermal cells. Negative controls were pSPYNE173/pSPYCE (M) and LlZFHD4-pSPYNE/LlDREB1-pSPYCE. Scale bars for LlZFHD4-pSPYNE/LlDREB1-pSPYCE, 100 μm; for others, 25 μm.

**Figure 7 ijms-20-03225-f007:**
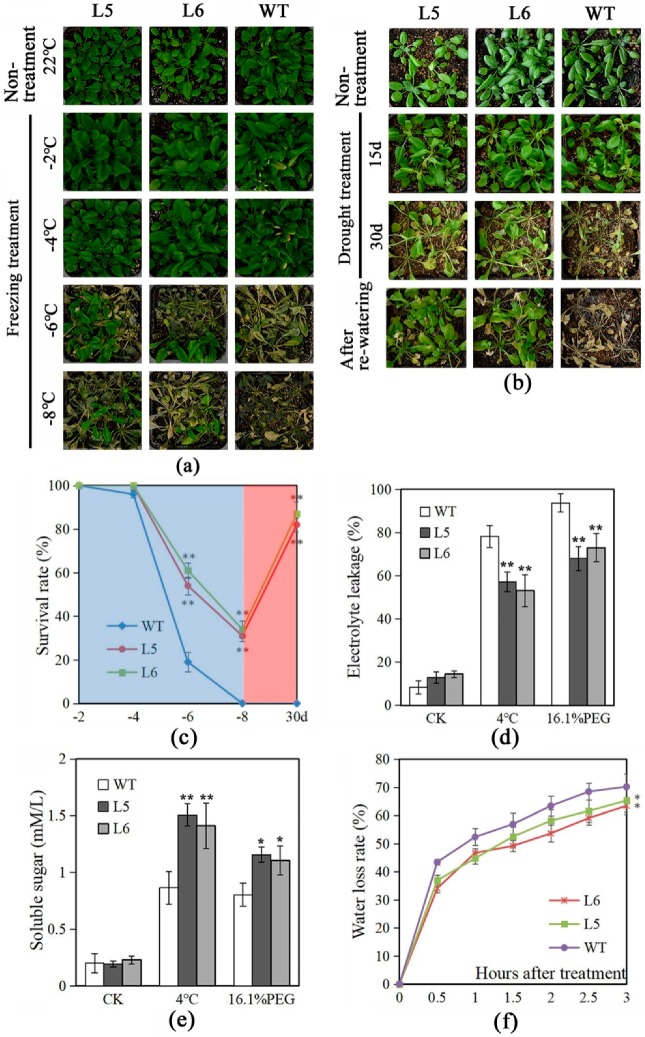
Overexpression of *LlNAC2* in Arabidopsis enhances plant tolerance to cold and drought stresses. Performance of WT and *LlNAC2* transgenic plants after freezing (**a**) and drought (**b**) treatments. (**c**) Survival rate of plants in (**a**) under freezing temperatures (blue region) and in (**b**) after drought treatment for 30 days (red region). Relative electrolyte leakage (**d**) and soluble sugar content (**e**) in WT and *LlNAC2* transgenic lines after 4 °C and 16.1% PEG6000 (−0.5 MPa) treatment for 3 h. (**f**) Water loss rate of leaves from WT and *LlNAC2* transgenic plants. Three biological replications were performed, each replication contains 30 plants. The bars show the SD. Asterisks indicate a significant difference 0.01 < * *p* < 0.05 and ** *p* < 0.01 compared with the corresponding controls.

**Figure 8 ijms-20-03225-f008:**
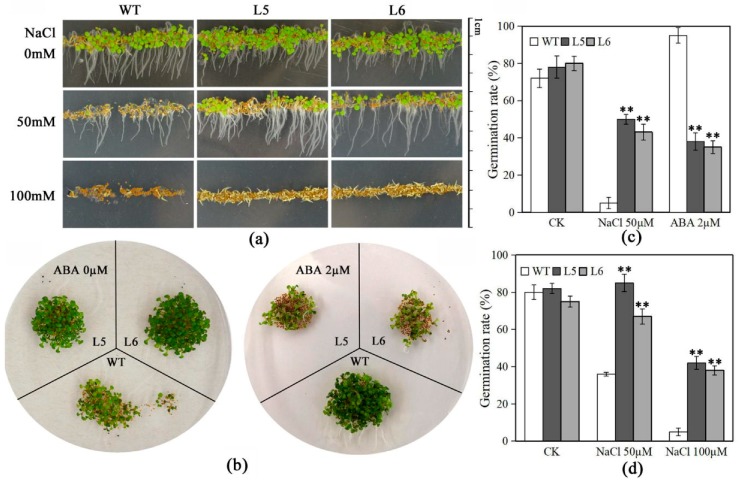
*LlNAC2* transgenic lines showed ABA hypersensitivity and enhanced salt tolerance. Germination of WT of Col-0 and 35S-LlNAC2 seeds on MS supplemented with (**a**) 50 mM or 100 mM NaCl and (**b**) 2 µM ABA assayed by both (**c**) cotyledon greening and (**d**) radicle protrusion. For (**a**) and (**b**), images were acquired after 7 days of incubation at 22 °C. Three biological replications were performed. The bars show the SD. Asterisks indicate a significant difference ** *p* < 0.01 compared with the corresponding controls.

**Figure 9 ijms-20-03225-f009:**
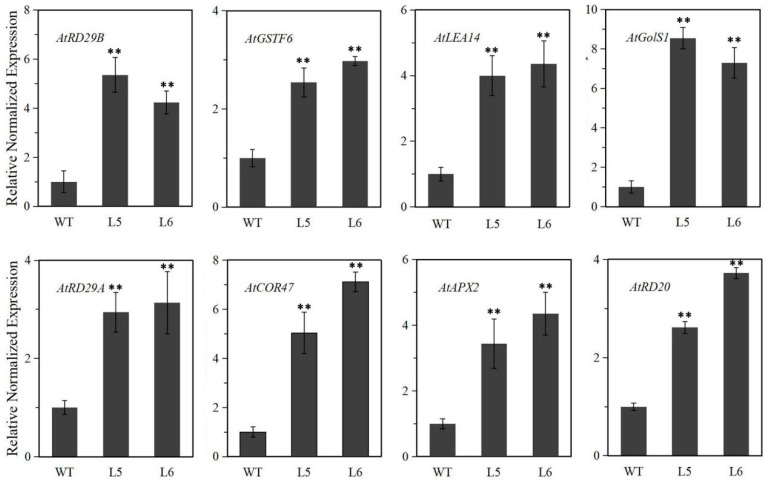
Transcript levels of the stress-related genes under normal condition in WT and *LlNAC2* transgenic Arabidopsis plants. Three biological replications were performed. The bars show the SD. Asterisks indicate a significant difference (** *p* < 0.01) compared with the corresponding controls.

**Figure 10 ijms-20-03225-f010:**
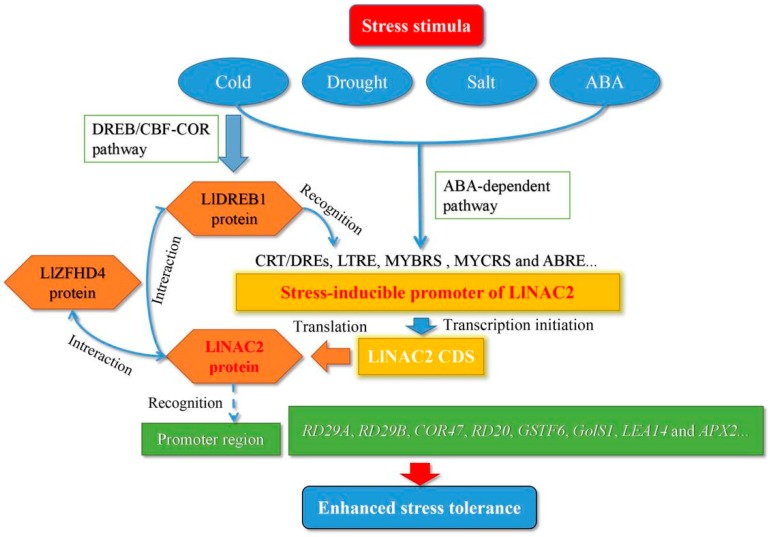
LlNAC2 is involved in both DREB/CBF-COR and ABA-dependent pathways to mediate stress tolerance of the tiger lily.

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
