# Peer review of "A Stress-Responsive NAC Transcription Factor from Tiger Lily (LlNAC2) Interacts with LlDREB1 and LlZHFD4 and Enhances Various Abiotic Stress Tolerance in Arabidopsis"

_ijms, 2019, doi:10.3390/ijms20133225_

Round 1

Reviewer 1 Report

Dear Authors The study is well designed and conducted Best of luck

Author Response

Reply to Reviewer 1

Thanks for your comments to this manuscript. The English language style, grammar and spelling in this manuscript have been carefully checked. We are so grateful for your commendation to this manuscript.

Best wishes to you!

Yingmin Lyu

Reviewer 2 Report

The paper describes a study on characterization of a cold-induced NAC transcription factor from tiger lily. The Authors designed a robust and extensive experimental approach to investigate the role of the LINAC2 in abiotic stress resistance, as well as identified some characteristic cis-acting elements involved in multiple stress response. Moreover, the interaction of LINAC2 promoter with L1DREB1 protein was proven by yeast one-hybrid assay. Furthermore, the overexpression of LINAC2 in Arabidopsis enhanced the toleration to cold and drought stresses. On the other hand, the transgenic lines showed increased sensitivity to ABA. Both transgenic Arabidopsis lines tested displayed increased expression of the stress-related genes under normal culture conditions.

The manuscript is well-written, only minor language editing is required. Both methodological section and the results were described extensively and in consistent manner, that raises no questions from my side. I would recommend to specifically state in the description of Figures if the SD was calculated for biological replicates, as “three independent experiments” may also mean plain technical replicates. The Materials and methods sections states that it indeed were biological replications, thus, it should not be a problem to state it in Figures’ legends as well. Presentation of the results is clear and meaningful, conclusions drawn are fully supported by the results of the experiments conducted.

Author Response

Reply to Reviewer 2

Thanks for your comments to this manuscript. We have revised the paper according to the suggestion: The “three independent experiments” were replaced by “three biological replications” in all Figure legends involving SD calculation. Also, the English language style, grammar and spelling in this manuscript have been carefully checked. We are so grateful for your commendation to this manuscript.

Best wishes to you!

Yingmin Lyu
